# Disruption of the Unique ABCG-Family NBD:NBD Interface Impacts Both Drug Transport and ATP Hydrolysis

**DOI:** 10.3390/ijms21030759

**Published:** 2020-01-23

**Authors:** Parth Kapoor, Deborah A. Briggs, Megan H. Cox, Ian D. Kerr

**Affiliations:** School of Life Sciences, University of Nottingham, Queen’s Medical Centre, Nottingham NG7 2UH, UK; parth.kapoor@nottingham.ac.uk (P.K.); deborah.briggs@nottingham.ac.uk (D.A.B.); megan.cox@nottingham.ac.uk (M.H.C.)

**Keywords:** ABC transporter, multidrug resistance, nucleotide binding domain, ATPase, pharmacology, molecular mechanism, structure

## Abstract

ABCG2 is one of a triumvirate of human multidrug ATP binding cassette (ABC) transporters that are implicated in the defense of cells and tissues against cytotoxic chemicals, but these transporters can also confer chemotherapy resistance states in oncology. Understanding the mechanism of ABCG2 is thus imperative if we are to be able to counter its deleterious activity. The structure of ABCG2 and its related family members (ABCG5/G8) demonstrated that there were two interfaces between the nucleotide binding domains (NBD). In addition to the canonical ATP “sandwich-dimer” interface, there was a second contact region between residues at the C-terminus of the NBD. We investigated this second interface by making mutations to a series of residues that are in close interaction with the opposite NBD. Mutated ABCG2 isoforms were expressed in human embryonic kidney (HEK) 293T cells and analysed for targeting to the membrane, drug transport, and ATPase activity. Mutations to this second interface had a number of effects on ABCG2, including altered drug specificity, altered drug transport, and, in two mutants, a loss of ATPase activity. The results demonstrate that this region is particularly sensitive to mutation and can impact not only direct, local NBD events (i.e., ATP hydrolysis) but also the allosteric communication to the transmembrane domains and drug transport.

## 1. Introduction

ABCG2 is one of at least three human ATP binding cassette (ABC) transporters that are capable of transporting multiple chemically diverse substrates out of cells. This polyspecificity can lead to multidrug resistance (MDR) of cancers, and it is well documented that ABCG2 expression is an indicator of poor prognosis in a number of hematological malignancies [1,2,3]. The broad substrate specificity of ABCG2 also underpins physiological roles in extra-renal urate transport [4], and disruption of ABCG2 urate transport function due to single nucleotide polymorphisms is a powerful predictor of serum hyperuricemia [5,6,7]. Additional physiological roles in stem cell protection, autophagy, and inflammatory disease further testify to the importance of understanding ABCG2 structure and mechanism [8,9,10,11].

ABCG2 is a half-sized ABC transporter, containing a single transmembrane domain (TMD) and a single nucleotide binding domain (NBD) in the polypeptide [12]. Transport function of the protein minimally requires the interaction of two ABCG2 monomers to form a dimer in order that NBD:NBD dimerization can result in the formation of two composite nucleotide binding pockets, as described in multiple other ABC proteins [13]. Indeed, numerous studies have documented the presence of ABCG2 dimers in the membrane, although this does not preclude the presence of functional larger assemblies [14,15,16,17].

In addition to being a half-sized transporter, ABCG2’s topology is also “reversed” compared to the ABCA-ABCD sub-families of ABC transporters. Whereas these other sub-families have an N-terminal TMD and a C-terminal NBD, ABCG2 (in common with the ABCG sub-family) has the opposite domain arrangement with the NBD at the N-terminus of the protein [12,18]. This altered domain orientation precluded the accurate modeling of ABCG2’s structure using other ABCs as templates, and it required the crystallographic structure of ABCG5/G8 and the cryo-electron microscopy structures of ABCG2 to be able to identify the structural implications of the reversed topology [19,20,21,22].

These structures have revealed that ABCG2 has an internal cavity for drug substrate and inhibitors that is connected to an external cavity and that the propulsion of bound substrates from one cavity to the other is likely to form the basis for transmembrane transport [19,20,21,23]. The structures have already provided the foundation for understanding numerous studies that have identified specific residues of importance in contributing towards drug specificity and inter-domain communication [24,25,26], and will continue to help understand how drug binding and transport is coupled to nucleotide binding and hydrolysis [27].

An unusual feature of the ABCG family structures (i.e. observed in both ABCG5/G8 and ABCG2) that appears restricted to this sub-family is the presence of two interaction interfaces between the NBDs. The first of these is the classical, well-defined interaction between the Walker A and B motifs of one NBD with the signature motif of the opposing NBD in order to generate the two sites for ATP catalysis [13,28,29]. The second, G-family specific interface, is located C-terminal to the “traditional” end of the NBD (i.e., C-terminal to the conserved His-switch motif [30]) and results in apparent constant contact of the ABCG family NBDs [19,20,21,22]. Among the residues located at this interface are those in a G-family specific sequence motif NPXDF [22] that have previously been identified as functionally significant in ABCG1 [31].

The function of this novel ABCG-specific NBD:NBD interface has hitherto been unexplored in ABCG2; in this study, we have mutated a selection of residues that are at this interface, both within and outside the NPXDF motif. Analysis of our mutants indicates that the interface is critical for ABCG2 trafficking and function.

## 2. Results

### 2.1. The ABCG Family Has A Novel NBD:NBD Interface

Recent structural data have demonstrated that ABCG family ABC transporters have a constant-contact NBD:NBD interface due to the presence of a conserved sequence of amino acids at the C-terminus of the NBD fold (Figure 1a; [22]). The presence of this ABCG-family specific interaction surface led us to hypothesize that disruption of this would be detrimental to ABCG2 function. We mutated several residues that are present at this interface, both within a recognized motif (NPXDF) and outside it (Figure 1b). Asp-292 forms inter-molecular interactions with the equivalent residue on the opposite NBD (Figure 1c) and so was mutated to either neutralize the charge (D292A) or reverse it (D292K). Tyr-247 (Y247), Asn-288, and Glu-285 were shown to be involved in a network of short-range (3.9–4.5 Å) electrostatic interactions across the NBD:NBD interface (Figure 1d,e). To disrupt these interactions, Y247 was mutated to alanine (Y247A), N288 was mutated to alanine to remove any polar interaction (N288A), or to aspartate to investigate the effect of introducing a charged residue at this position (N288D), and E285 was mutated either to remove the charge (E285A) or to reverse the charge (E285K).

### 2.2. The ABCG2 Mutation N288D Fails to Traffic to the Plasma Membrane

All residue mutations were constructed in an isoform of ABCG2 that is superfolder green fluorescent protein (sfGFP)-tagged to enable localization of the protein and to allow us to normalize functional data by the relative expression level [24]. Following the selection of stable cell lines, we conducted complementary investigations to reassure ourselves that mutant versions were being expressed effectively at the cell surface. Qualitatively, we imaged the stable cell lines on a confocal imaging plate reader visualizing the GFP signal from the sfGFP-ABCG2 fusion proteins. For most transfected cell lines, there was a clear membranous signal in the vast majority of cells (Figure 2), with the exception of N288D, which showed some membrane localized GFP fluorescence in addition to a broader intracellular fluorescence in a significant number of cells (Figure 2g). 

To further investigate cell surface localization, particularly in mutant versions where some intracellular protein expression was suspected (e.g., N288D), we employed a more quantitative method to complement the confocal imaging. A two-channel flow cytometry assay was used, in which we determined total GFP fluorescence as a marker for overall sfGFP-ABCG2 expression in parallel with the determination of 5D3 antibody reactivity, as this antibody recognizes a surface localized epitope of ABCG2. Quadrant plot analysis of the data for untransfected HEK293T (Figure 3a) demonstrated that over 99% of cells were both negative for GFP fluorescence and for 5D3 reactivity (Figure 3a, 99% of cells in lower left quadrant). By contrast almost 100% of cells expressing wild type (WT) sfGFP-ABCG2 were positive for both GFP and for 5D3 reactivity (Figure 3b, upper right quadrant). A similar pattern to WT was observed for all mutants investigated (a selection of which are shown in Figure 3c–e) with the notable exception of N288D (Figure 3f), which showed that a large proportion of the GFP-fluorescent cells did not have a significant cell-surface 5D3 reactivity, indicating that sfGFP-ABCG2-N288D was expressed but was improperly targeted to the cell membrane. This combination of qualitative and quantitative assays was used to exclude N288D from further analysis, as it appears that this residue mutation has altered protein stability. 

### 2.3. Drug Transport Is Altered in NBD Interface Mutants

Having established that the majority of mutants were still expressed and targeted correctly to the plasma membrane, we investigated their drug transport function using a flow cytometry assay as previously described [24]. In this assay, the inhibitor Ko143 [32] is employed alongside three fluorescent substrates in order to quantify the ABCG2-specific transport ability of the mutants. The GFP signal of each cell line is then used to normalize any drug transport effects for different levels of expression of the sfGFP-ABCG2 versions. The three drugs employed in the assay were (i) the anthracenedione mitoxantrone, which is a topoisomerase II inhibitor and a well-characterized transport substrate for ABCG2 [33]; (ii) the chlorophyll metabolite pheophorbide A, which is also a transport substrate for ABCG2 [34]; and (iii) daunorubicin, another topoisomerase II inhibitor that is not a transport substrate for WT ABCG2, but is transported by ABCG2 mutants, notably R482G [33,35].

Negative controls (cells lacking ABCG2 expression) displayed no Ko143-inhibitable transport of any of the three drugs used (e.g., Figure 4, left hand column and Table 1). Cells expressing an ABCG2 containing the Walker B neutralizing mutation (E211Q) in both NBDs (which prevents cycles of ATP hydrolysis [36]) also showed no transport of the three drugs (Table 1), despite this mutation being effectively localized to the cell surface (see Figure 3c). Cells expressing wild type ABCG2 behaved as previously demonstrated, being capable of exporting mitoxantrone and pheophorbide in a Ko143-dependent manner, whilst showing no transport of daunorubicin (Figure 4, 2^nd^ panel; Table 1). The mutation E285K maintained Ko143-inhibitable transport of mitoxantrone and pheophorbide; this mutation also showed subtle but statistically significant (Table 1; Figure 4, 3^rd^ panel) transport of daunorubicin. The starkest phenotype was observed for mutations at position 292. The charge reversal substitution (D292K; Figure 4, right hand panel) completely abrogated transport of both mitoxantrone and pheophorbide A.

Quantification of the normalized transport assay data for all mutants (see methods and [24]) demonstrated that two mutants (E285A and N288A) had transport parameters that were indistinguishable from wild type ABCG2 (Table 1). Mutant Y247A showed enhanced transport of pheophorbide A compared to the wild type (mean fractional differences of 5.63 vs. 2.45; Table 1) but not for mitoxantrone. Mutant E285K displayed increased transport for all three drugs, including daunorubicin, for which the relative efflux was 0.26 compared to the positive control for daunorubicin transport (R482G, 0.53; Table 1). Both mutations at position D292 resulted in a complete loss of transport activity with values indistinguishable from untransfected HEK cells and the E211Q catalytically inactive mutant (Table 1). In summary, mutation at the NBD:NBD interface that is unique to the ABCG family results in a spectrum of changes to transport, including alterations to specificity (E285K), alterations to capacity (E285K and Y247A), and loss of function (D292A or D292K).

### 2.4. Abrogation of Drug Transport in D292 Mutants Is Due to Inhibition of ATPase Activity

The transport phenotype of the D292 mutant isoforms was very similar to that of the E211Q mutation, which is known to be defective in ATP hydrolysis. One obvious reason for the failure of transport by D292A and D292K mutants would be a similar lack of ATP hydrolysis by these two mutant isoforms. Whole cell membrane preparations were prepared from large-scale cell cultures of these two mutants and appropriate controls. The ATPase activity of ABCG2 is notoriously difficult to measure due to the presence in membranes of endogenous stimulators of the activity (e.g., cholesterol). For this reason, we measured inorganic phosphate release in the presence of the ABCG2 substrate Lucifer yellow (as mitoxantrone interferes with the colorimetric determination of the phosphomolybdate complex) from ABCG2-containing membranes and identified the fraction of this release that was both ATP-dependent and Ko143-inhibitable. Membrane preparations from different cell lines displayed different levels of this activity (Figure 5), so the justifiable comparison is to determine which mutants had a Ko143-inhibitable activity, rather than comparing one mutant’s activity with another. For this reason, we were unable to show whether E285K had an elevated activity. Of the isoforms tested, only the WT version of the protein displayed a Pi release that was inhibited by Ko143 (Figure 5, *p* < 0.05). The well characterized catalytically inactive mutant, and the two new NBD interface mutants failed to show any Ko143 inhibition of Pi release, confirming that D292A and D292K mutations prevent ATP hydrolysis by ABCG2, resulting in abrogation of transport in cell-based studies (Figure 5).

## 3. Discussion

Structural data on the ABCG family have brought us considerably further forwards in understanding the mechanism of these half-transporters [18,23]. Until there were structural data, the region between the NBD of ABCGs and the first transmembrane (TM) helix (over 150 residues in total, e.g., from ca. residue 240 to 390 in ABCG2) was very poorly understood. The advances made in crystallographic and cryo-electron microscopy analysis of ABCG5/G8 and ABCG2 has shed much light on this region with the demonstration of a “connecting helix” [22] immediately preceding the TMD and an unexpected additional NBD:NBD contact that results in constant contact of ABCG family NBDs [19,20,21,22]. This is dissimilar to the NBD interface of ABCB transporters where ATP binding seems to be concomitant with NBD dimerization. The novel G-family specific NBD:NBD interface is extensive and includes residues in a 50 amino acid sequence (from ca. 245–295 in ABCG2). Within this region is a G-family conserved motif (NPXDF; residues 289–293 in ABCG2), but analysis of the interface identifies several other residues localized here that are involved in short range cross-interface interactions. In this study, we analysed several residues located at this interface and demonstrated effects on protein targeting, drug transport, and ATPase activity.

Of the residues we analysed, one, namely N288D, was shown to have a dramatic effect on cell surface localization with only 15% of cells expressing this mutant on the cell surface. Additional confocal microscopy on fixed cells indicated that the protein was trapped in a cytoplasmic compartment, most likely the endoplasmic reticulum (Appendix A), indicating that this residue was not being trafficked correctly. Similar effects on protein localization have been shown for mutations in the glycosylated region of the protein (extracellular loop 3; [37,38]) as well as with the Q141K polymorphism in the NBD:TMD interface. It is thus clear that destabilization of ABCG2’s trafficking can come via direct effects on the glycosylation, which is necessary for trafficking, or via indirect, allosteric effects. The destabilization of the NBD:NBD interface is probably the result of introducing two acidic groups (as ABCG2 is a dimer all our mutations introduce two amino acid changes into the ABCG2 dimer) very close to the NPXFD motif. Indeed, mutations of the adjacent residue (also Asn) in ABCG1 resulted in impaired trafficking and function when the mutation was Asn → Asp [31].

The importance of this interface in protein dynamics was evidenced by some mutations having a gain-of-function in transport assay experiments. E285K had a higher relative transport of both mitoxantrone and pheophorbide A; remarkably this mutant, which is far from the TMDs also conferred a slight, but significant transport of the non wild-type ABCG2 substrate daunorubicin. The only other residues to be mutated to date which confer daunorubicin transport are at position 482 in TM3 [33,39] and at positions 523 in TM5 and 640 in TM6 [24]. That several positions in ABCG2 (at distinct sites on the protein) can confer altered specificity suggests that substrate specificity of ABCG2 can be influenced by changes outside of the cavities identified as substrate/inhibitor binding sites [19,20]. The observation that E285K and Y247A (which also had some altered transport activity) lie outside the NPXDF motif confirms that this motif—whilst highly conserved in the ABCG family—is not the sole contributor to the constant contact interface.

Within the NPXDF motif itself, we only mutated the aspartate; both these mutations were defective in transport activity, although both were expressed at the cell surface. This is different from the situation with ABCG1, where mutation of the equivalent residue to alanine did not affect cholesterol transport [31]. We were unable to detect a Ko143-sensitive ATPase activity of these mutations, indicating that residues at this interface can exert important effects on the binding and hydrolysis of nucleotides.

The ABCG family NBD-TMD region has been a region that has been investigated relatively infrequently [31,40], perhaps because it lacks homology to other ABCs and because until recently there have been no structural data for this region. This hampers both hypothesis-driven research and any structure-based interpretations of results. With several single nucleotide polymorphisms (SNPs) located in this area (G268R, P269S, and D296H), it indicates that more research into the effects of these SNPs on drug export by ABCG2 as well as the function of the altered ABCG2 proteins is needed [5]. Finally, even with the recent structural advances there are still several stretches of amino acids whose structures are unresolved, suggesting a significant degree of conformational plasticity in this region. Our data suggest that the NBD constant contact interface is a region of the transporter that contributes to both ATP hydrolysis and NBD:TMD communication and warrants further structural and functional elucidation.

## 4. Materials and Methods

### 4.1. Site Directed Mutagenesis

All single mutations were made in a vector (p31zeo_sfGFP_ABCG2) that encodes an N-terminal superfolder-GFP tagged ABCG2, as previously described [14]. Mutations were introduced using oligonucleotide-directed site-directed mutagenesis (primers listed in Appendix A), with Pfu Polymerase (Promega, Southampton, UK). Parental DNA was removed by DpnI digestion, and putative mutant plasmids were obtained following transformation of chemically competent DH5α cells and commercial plasmid preparation kits (Machery-Nagel, Loughborough, UK). Confirmation that only the required mutation was present was obtained by DNA sequencing across the entire sfGFP-ABCG2 cDNA (Source Bioscience, Nottingham, UK).

### 4.2. Cell Culture

Human embryonic kidney cells (HEK293T) were maintained in T25 flasks (Corning) at 37 °C, 5% CO_2_ in Dulbecco’s Modified Eagle Medium (DMEM, 4500 mg/*L*-glucose, *L*-glutamine, sodium pyruvate, and sodium bicarbonate) supplemented with 10% (*v*/*v*) fetal calf serum (Sigma, Gillingham, UK), 100 units/mL penicillin, and 100 µg/mL streptomycin (Sigma, Gillingham, UK). The cells were passaged at 90% confluence by trypsinization.

### 4.3. Transfection and Selection of Stable Cell Lines

Cells were seeded at 1.5–2 × 10^5^ cells/well into a 6-well plate 24 h prior to transfection. Three hours prior to transfection, the media were replaced with DMEM supplemented with 5% (*v*/*v*) FCS. Cells were transfected by adding preformed linear polyethyleneimine (PEI; Polysciences Inc., Mannheim, Germany) and DNA complexes (molar PEI nitrogen/DNA phosphorous ratio of 15:1) [41]. Successful transfection was confirmed 24 h later using an inverted epifluorescence microscope (Hg lamp, Carl Zeiss, Cambridge, UK), and the media were then replaced with DMEM supplemented with 10% (*v*/*v*) FCS. Another 24 h later, cells were transferred to T25 flasks and media supplemented with 200 µg/mL Zeocin (Thermo-Fisher Scientific, Loughborough, UK). Zeocin-resistant colonies of transfected cells were developed over a 14–21 day period. Longer-term passaging and the maintenance of stable cell lines were carried out at a reduced Zeocin concentration (40 µg/mL).

### 4.4. Cell Imaging

Live cell imaging of HEK293T cells stably transfected with sfGFP_ABCG2 mutant isoforms was performed using an ImageXpress (IX) Ultra confocal plate reader (Molecular Devices, Wokingham, UK), using a plan-apochromat 40× objective, with an excitation wavelength of 488 nm and an emission bandpass filter of 525/50 nm. Cells were seeded in poly-L-lysine coated, 96-well black-walled, clear-bottom plates (Greiner) at a cell density of 3 × 10^4^ cells/well in DMEM 24 h before imaging. Cells were subsequently washed twice with pre-warmed (37 °C) phenol-red free HBSS (Hank’s Balanced Salt Solution, Sigma, Gillingham, UK) immediately prior to imaging.

### 4.5. Drug Transport Analysis

Drug accumulation assays were performed as previous described [24]. Briefly, cells were seeded at 1 × 10^6^ cells/mL in phenol-red free DMEM and incubated with either DMSO (solvent control 0.2% *v*/*v*), mitoxantrone (MX, 10 µM), pheophorbide A (PhA, 10 µM), or daunorubicin (DNR, 10 µM) in the presence or absence of the ABCG2 inhibitor Ko143 (1 µM, [32]) at 37 °C for 30 min with occasional agitation. Excess drug was removed by centrifugation prior to a second incubation (37 °C, 60 min) with either phenol-red free DMEM only or phenol-red free DMEM plus Ko143. Wavelength pairs for detection of expression (GFP) or transport (MX, PhA, DNR) were 488/526 nm, 635/670 nm, 355/692 nm, and 490/630 nm, respectively. Daunorubicin fluorescence was separated from GFP fluorescence during data acquisition by compensation.

Data were analysed using Kaluza analysis version 1.5 (Beckman Coulter, High Wycombe, UK) as previously described [24]. Briefly, cells were gated based on size, dispersity, and ABCG2-expression (GFP fluorescence). The fractional difference in vehicle-subtracted, median drug fluorescence between samples with drug plus inhibitor compared to samples of the drug alone was calculated. This was corrected for the median expression level of ABCG2 isoforms compared to wild type GFP-ABCG2. The normalized values were analysed using GraphPad Prism and were subjected to one-way ANOVA with a Dunnett’s multiple comparisons against wild type ABCG2 to determine if any of the mutations differed in their ability to efflux drug.

### 4.6. Cell Surface Expression Analysis

Cells were seeded at a cell density of 1 × 10^6^ cells/mL in a blocking buffer (PBS containing 1% *w*/*v* BSA) and incubated with primary monoclonal antibody anti-ABCG2, clone 5D3 (1:200; Millipore), or an isotype control. Cells were incubated on ice for 30 min and subsequently washed by two cycles of pelleting (350 g, 5 min, 4 °C) and resuspension in blocking buffer. Cells were then incubated with secondary antibody AlexaFluor647 (AF647; 1:200 Thermo-Fisher Scientific, Loughborough, UK) on ice for 60 min, followed by the same washing steps. Finally, the cells were resuspended in blocking buffer and analysed for GFP (488/526 nm) and AF647 (650/670 nm) using an Astrios cytometer (Beckman Coulter, High Wycombe, UK). Data were analysed using Kaluza, with prior gating for size and dispersity.

### 4.7. Membrane Preparation and ATPase Assay

HEK293T cells expressing the isoforms of interest were pelleted and washed to remove excess media. Cells were resuspended in 10 mM Tris pH 7.4, 0.25 M sucrose, and 0.2 mM CaCl_2_, supplemented with protease inhibitor cocktail set III (Merck) and lysed by two cycles of nitrogen cavitation (1000 psi, 15 min 4 °C, Parr Instruments, Moline, Illinois, USA). Cell debris was removed by centrifugation (1500× *g*, 20 min, 4 °C), and whole cell membranes were subsequently pelleted from the supernatant by ultracentrifugation (100,000× *g*, 60 min, 4 °C). Membranes were resuspended in 50 mM Tris, 250 mM sucrose, and 150 mM NaCl, pH 8.0 (supplemented with protease inhibitors) by shearing 20–30 times through a 25 G needle. The protein concentration in membranes was measured by the Lowry assay (BioRad DC protein assay, Watford, UK).

The ATPase activities of ABCG2 in HEK293T membranes were measured using a colorimetric assay to monitor the release of inorganic phosphate, based on the method described by [42]. All reagents were from Sigma (Gillingham, UK) or Thermo-Fisher Scientific (Loughborough, UK). ATPase measurements were performed in 96-well clear-bottom plates with membranes containing the ABCG2 isoform of interest (100 µg protein) in a final assay buffer containing 5 mM MgSO_4_, 50 mM KCl, 2.5 mM EGTA, 1.3 mM ouabain, and 2 mM DTT. Reactions were supplemented with either solvent control DMSO (1% *v*/*v*) or substrate lucifer yellow (100 µM) in the presence or absence of 1 µM Ko143. The reaction was equilibrated at 37 °C for 5 min. Each reaction condition was then completed (to a final reaction volume of 50 µL) by the addition of either 5 mM ATP or 5 mM AMP (from buffered stock solutions in 50 mM Tris, 250 mM sucrose, and 150 mM NaCl, pH 8.0) and incubated at 37 °C for 15 min. The reaction was stopped with SDS (2.5% final concentration). Phosphate was detected by sequential addition of distilled water (50 µL), phosphate detection reagent (40 µL; 1% (*w*/*v*) ammonium molybdate, 0.014% (*w*/*v*) antimony potassium tartrate, 2.5 N sulfuric acid), and ascorbic acid (20 µL; 1% *w*/*v*). The plate was incubated at room temperature for 12 min and the absorbance intensity from the formation of phosphomolybdate was measured at 880 nm using a SpectraMax^®^ M2 microplate reader (Molecular Devices, Wokingham, UK). The data were analysed using Prism 7 (GraphPad, San Diego, USA). The data are reported as ATP-dependent inorganic phosphate release in the presence or absence of Ko143.

### 4.8. Data Analysis

Other numerical analyses were performed using Microsoft Excel and statistical analysis were performed using GraphPad Prism, using ANOVA with Dunnett’s post-test for multiple comparisons between mutant versions of ABCG2 (drug transport data), or unpaired *t*-tests for pairwise comparisons (ATPase in the presence and absence of Ko143).

## Figures and Tables

**Figure 1 ijms-21-00759-f001:**
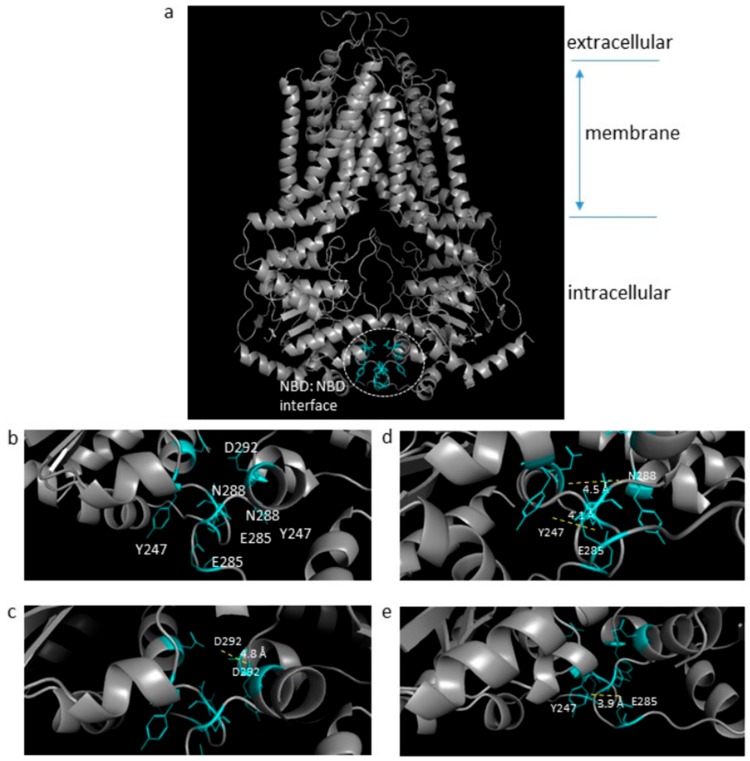
Structural mapping of the highly conserved residues at the NBD:NBD interface. The ABCG2 homodimer cryo-EM structure (**a**, whole view) with the enlarged view of the NBDs showing the novel constant interface (blue, **b**). Structural mapping of the mutated NBD interface residues (Y247, E285, N288, and D292) are highlighted as cyan sticks (**c**–**e**) depicting the relative positions of the possible interaction residues (<5 Å) at this interface. Figure generated on PyMOL software using PDB: 6ETI; (Jackson et al., 2018) [20].

**Figure 2 ijms-21-00759-f002:**
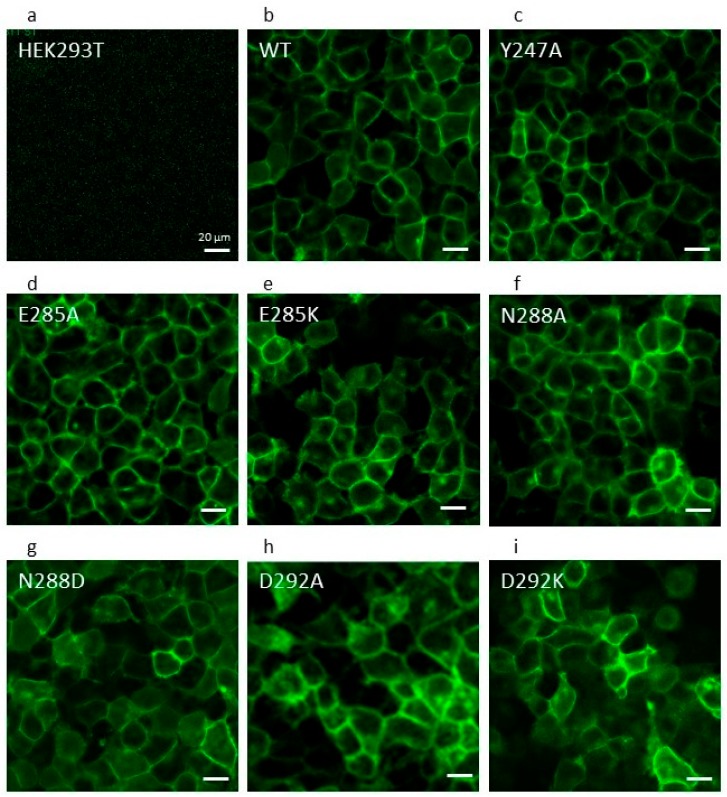
Cell surface localisation of NBD interface residues. Confocal microscopy was performed to visualize cell surface localization of ABCG2 isoforms in stable cell lines (**a**–**i**). The majority of the mutant isoforms showed evidence of cell membrane localization with varying levels of protein expression comparable to the wild type. Live cells were imaged using an ImageXpress (IX) Ultra confocal plate reader (plan-apochromat 40× objective, with excitation wavelength of 488 nm and emission bandpass filter of 520/50 nm). Scale bars represent 20 µm.

**Figure 3 ijms-21-00759-f003:**
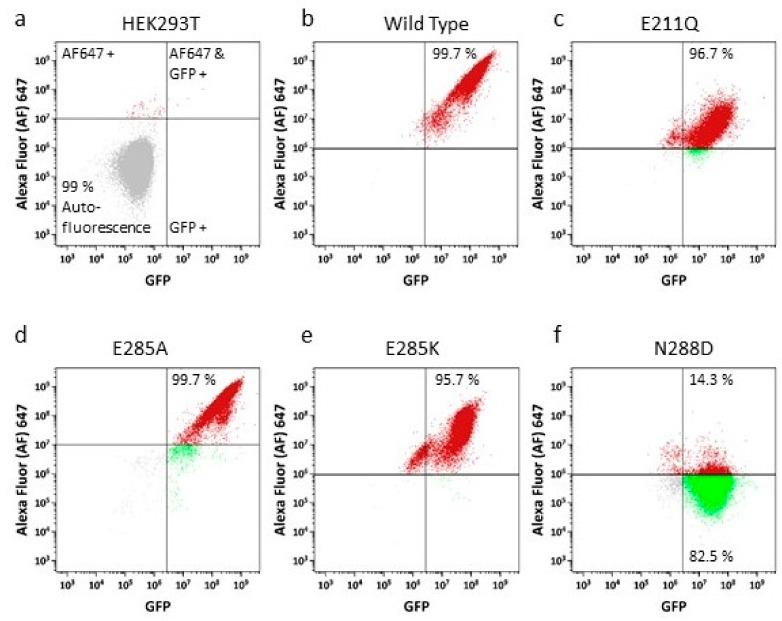
Cell surface expression for NBD interface mutant isoforms using a two channel flow cytometry. The cells were labelled with 5D3 antibody and a secondary antibody conjugated to AlexaFluor (AF) 647 enabling individual cells to be quantified for cell surface ABCG2 expression (y-axis) and total GFP expression (x-axis). The cells were analysed for GFP and AF647 expression using an Astrios flow cytometer. The cells in the upper right corner of the quadrant represent the cells with ABCG2 surface expression. The percentage of cell surface expression for HEK293T (**a**), wild type (**b**), catalytically inactive control (E211Q, **c**) and NBD interface mutant isoforms (**d**–**f**) have been documented in the quadrant.

**Figure 4 ijms-21-00759-f004:**
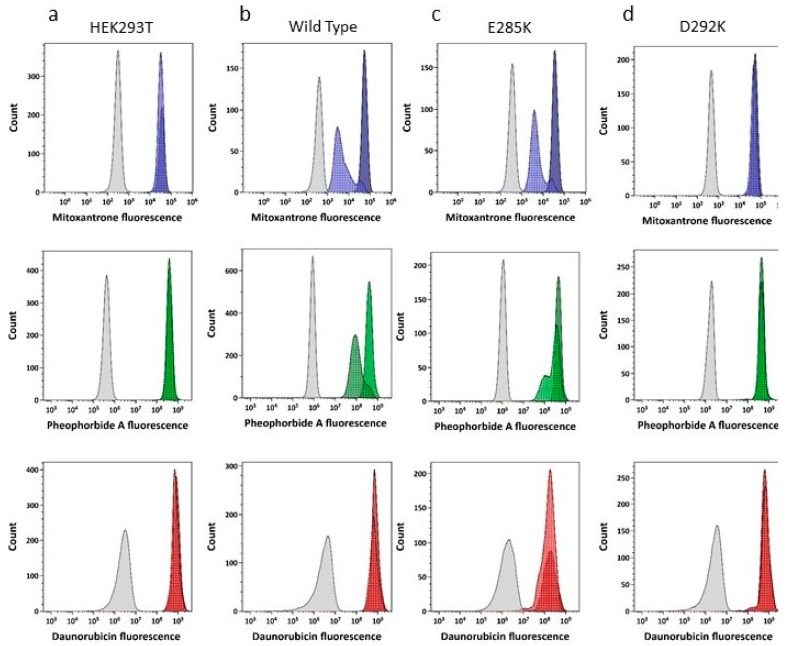
Altered drug transport function in ABCG2 NBD interface mutants. Single channel histogram plots to quantify mitoxantrone (top row), pheophorbide A (middle panels) and daunorubicin (lower panels) fluorescence in HEK293T cells (**a**), cells expressing wild type ABCG2 (**b**), and two NBD interface mutants (E285K, (**c**) and D292K, (**d**)). Grey histograms represent autofluorescence of cells. The dotted blue, green and red histograms depict equilibrium level of substrate accumulation (mitoxantrone, pheophorbide-a and daunorubicin respectively). The effects of inclusion of Ko143, a specific ABCG2 inhibitor, are shown in the solid colour histograms.

**Figure 5 ijms-21-00759-f005:**
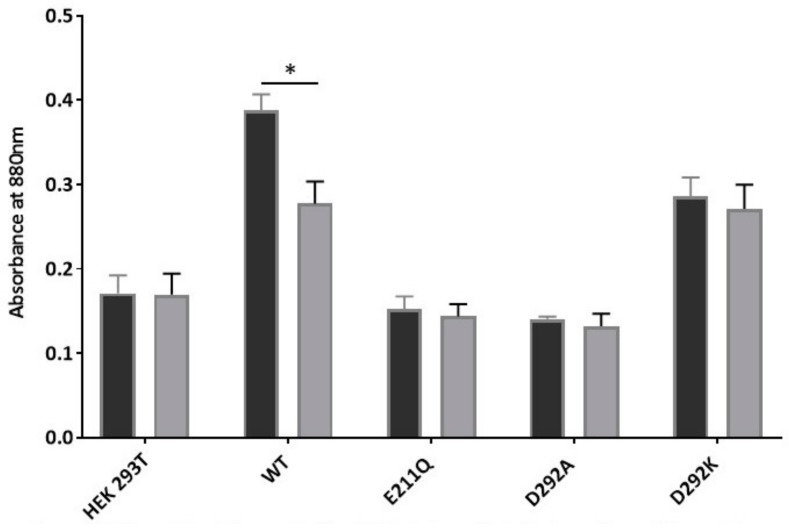
ATPase activity of transport-inactive NBD interface mutants. Crude membranes (20 μg protein) were incubated with lucifer yellow (100 μM; dark bars bar) in absence or presence (light bars) of Ko143 (1 μM). The results show that ATP-specific Pi measured by colorimetric determination of phosphomolybdate complexes. Only WT ABCG2 demonstrates a level of Pi release which is inhibited with Ko143 (* *p* < 0.05), demonstrating ABCG2 specific Pi release, confirming that D292A and D292K are ATPase deficient mutants.

**Table 1 ijms-21-00759-t001:** Ko143-inhibited transport of fluorescent drugs by NBD interface mutant isoforms. The table depicts the mean (with standard error of the mean) of the fractional difference in Ko143-inhibited drug efflux compared to the wild type. The raw data for mitoxantrone and pheophorbide A were processed using one-way ANOVA and a post statistical test (Dunnett’s multiple comparison) against the wild type. The data for the daunorubicin efflux were subjected to log transformation due to positive skewed data for R482A mutant isoform, prior to analysis of significance. The data are representative of *n* ≥ 3 independent repeats. Asterisks indicate the level of significance with *p* ≤ 0.05 for * and *p* ≤ 0.01 for ** compared to wild type ABCG2.

Cell Line	Mitoxantrone	Pheophorbide A	Daunorubicin
HEK293T	0.21 ± 0.07	−0.07 ± 0.05	0.03 ± 0.01
WT-ABCG2	11.90 ± 1.88	2.45 ± 0.48	0.03 ± 0.03
E211Q	1.22 ** ± 0.20	−0.05 ** ± 0.07	−0.02 ± 0.08
Y247A	12.19 ± 1.18	5.63 ** ± 0.05	0.20 ± 0.05
E285A	8.51 ± 1.53	3.05 ± 0.88	0.08 ± 0.03
E285K	20.26 ** ± 2.85	7.20 ** ± 1.06	0.26 ** ± 0.06
N288A	11.53 ± 2.40	4.37 ± 0.17	0.19 ± 0.04
D292A	1.46 ** ± 0.90	0.13 * ± 0.10	0.12 ± 0.04
D292K	0.81 ** ± 0.07	0.10 * ± 0.17	0.05 ± 04
R482A			0.53 ** ± 0.05

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
