# Peer review of "Disruption of the Unique ABCG-Family NBD:NBD Interface Impacts Both Drug Transport and ATP Hydrolysis"

_ijms, 2020, doi:10.3390/ijms21030759_

Round 1

Reviewer 1 Report

Disruption of the unique ABCG-family NBD:NBD interface impacts both drug transport and ATP hydrolysis

The present manuscript reports the effect of mutations in 4 amino acids located in a short strecht peripheral to the Nucleotide Binding Domain in the ABC-transporter encoded by the human gene ABCG2.
While the manuscript reports sound results on the effect of these mutations, the overall impression is that the present study is still too preliminary for publication and not sufficiently clear how it contributes to the understanding of ABC transporters. The effects of mutations are clear but the physiological or biochemical significance is not pursued; also, no molecular mechanism that could explain in detail the observed results is provided. As a consequence, the Discussion section at times shows a repetition of the results already described in the previous section. Furthermore, the main experiments fail short to provide an indepth view on the effect of the mutations. For example, they should be complemented with structural or kinetic determinations that could provide more information confirming whether, to which extent and in which direction the mutations result in the predicted disorganisation of the nearby NBD. Also, experiments on cellular effects on resistance to antitumour drugs, or association with genetic variants found in the human genome, could also be used to complement this work.

Author Response

We thank the reviewer for their comment that the "effects of the mutations are clear". We recognise that mechanistic explanation to links the residues to allosteric or structural effects on the protein is an important aspect, but we would argue that this would be a follow-up piece of work. We have constructs that enable us to examine protein conformation using for example bioluminescence energy resonance transfer and fluorescence correlation spectroscopy. These experiments are very time-consuming and out intention is to select favoured mutations from this study and our other recent study (Cox, 2018) to take forward an examine mechanistically. 

Reviewer 2 Report

Dear Authors,

Your paper “Disruption of the unique ABCG-family NBD:NBD interface impacts both drug transport and ATP hydrolysis” was really interesting to read and fills an important but empty section of the understanding of the ABCG-family and particularly ABCG2. I have only a few concerns and recommendations, please find it below.

Concerns:

1. The 5D3 antibody is binding in a conformational dependent manner (DOI: 10.1074/jbc.M411338200). With wildtype ABCG2 this is not a problem but with a mutant that potentially could fix one or the other conformation can lead to a biased outcome. I think this should be addressed as in my opinion the N288D mutant is as good on the cell surface as the others (Fig 2 g) while the flow cytometry shows a marked change.

2. Interestingly, the E285K mutant showed increased transport capability. It would be interesting to include it in the ATPase activity assay. Especially because while the E211Q and D292A mutants are seemingly inactive the D292K mutant has a wildtype level ATPase activity which is not inhibitable by Ko143. Just as the E285K mutant can gain transport capability, the same way another mutant can gain immunity against an inhibitor.

3) line 114: typo in ‘mutations’ (should be mutants)

Suggestions:

Figure 1.: Using a white background and highlighting with dark colors gives a nicer impression to the image.

Figure 4.: Coloured histograms are hard to differentiate, please use different colors instead of a pattern.

Author Response

The 5D3 antibody is binding in a conformational dependent manner (DOI: 10.1074/jbc.M411338200). With wildtype ABCG2 this is not a problem but with a mutant that potentially could fix one or the other conformation can lead to a biased outcome. I think this should be addressed as in my opinion the N288D mutant is as good on the cell surface as the others (Fig 2 g) while the flow cytometry shows a marked change.

We thank the reviewer for this insightful comment. The point we are making with N288D is that this isoform shows a considerable intracellular retention - this is seen both with the live cell imaging with GFP and with the flow cytometry using 5D3. We are aware of the 5D3 antibody's conformational sensitivity although note that in the structural biology work of Kaspar Locher the protein conformation was very consistent in the presence or absence of 5D3. We have carried out western blots with BXP-21 and BXP-53 antibodies but didn't include these in the manuscript as both those antibodies are raised against an epitope that includes the C-terminus of the NBD under investigation and so these would also have needed the cautionary note that the antibody reactivity could have been influence by the mutation. We have included as a supplementary figure in the revised manuscript further data on fixed cells that confirms that intracellular localisation of this mutant, thereby justifying its exclusion from the functional assay. 

2. Interestingly, the E285K mutant showed increased transport capability. It would be interesting to include it in the ATPase activity assay. Especially because while the E211Q and D292A mutants are seemingly inactive the D292K mutant has a wildtype level ATPase activity which is not inhibitable by Ko143. Just as the E285K mutant can gain transport capability, the same way another mutant can gain immunity against an inhibitor.

We thank the reviewer for this comment as well. The ATPase measurements were made in cell membranes and so contain a significant number of other ATPases which will all contribute to the signal measured. They are the principal reason why different membrane preps show different levels of Pi release. We can identify the ABCG2 specific component by using Ko143 to inhibit ABCG2 (at the concentration we use Ko143 is ABCG2 specific). This enables us to determine a catalytically inactive mutant very easily as there is no ABCG2-specific Ko143 inhibition. However, for mutants were elevated activity is suspected this is trickier due to the high and variable background ATPases, and the inability to further enhance this with ABCG2 specific substrates. To study E285K ATPase activity would require us to purify the protein and currently the SMALP technique we used for purification does not allow ATPase activity due to its acute Mg sensitivity (see e.g. https://onlinelibrary.wiley.com/doi/full/10.1002/anie.201610778 for discussion of this). We are trying to circumvent this so that we can determine E285K's ATPase activity but currently this is not yet possible.

3) line 114: typo in ‘mutations’ (should be mutants)

Suggestions:

Figure 1.: Using a white background and highlighting with dark colors gives a nicer impression to the image.

Figure 4.: Coloured histograms are hard to differentiate, please use different colors instead of a pattern.

We have amended the manuscript to take account of these useful suggestions

Reviewer 3 Report

The manuscript by Kapoor et al. is a very interesting study on the role of the NBD:NBD interface in the function/structure relationships of the ABCG transporters. The manuscript is well done both in terms of experimental design and data interpretation. The discussion of results should be improved. Some concerns are listed below.

1) The mutation N288D is deleterious in terms of protein trafficking to the plasma membrane. Some more discussion should be added on this issue. Why mutation to Ala has not the same effect. Are glycosylated moieties of the protein involved in trafficking? May have N288D mutation a long distance effect on the protein site important for trafficking?

2) May performed mutations contribute in explaining functional variation due to known polymorphisms of ABC transporters?

3) Fig. 1 A. Please indicate the relationships of the transporter with the membrane and the inner/outer sides of the transporter; non expert readers will benefit from such drawing.

4) E285K seems to have transport capacity better than WT. This data should be commented. Which could be the explanation? What about ATP hydrolysis property of this mutant?

Author Response

The manuscript by Kapoor et al. is a very interesting study on the role of the NBD:NBD interface in the function/structure relationships of the ABCG transporters. The manuscript is well done both in terms of experimental design and data interpretation. The discussion of results should be improved. Some concerns are listed below.

1) The mutation N288D is deleterious in terms of protein trafficking to the plasma membrane. Some more discussion should be added on this issue. Why mutation to Ala has not the same effect. Are glycosylated moieties of the protein involved in trafficking? May have N288D mutation a long distance effect on the protein site important for trafficking?

We have amended the manuscript to include some limited speculation about how a long-distance effect on trafficking could be explained but we are cautious as this would require some much more detailed experimentation to confirm this. The role of glycosylation on ABCG2 trafficking is not entirely clear. Some authors report no effect on localisation (e.g. Diop 2005) but others report that defective glycosylation is associated with stability and trafficking defects (Nakagawa 2009, Haider 2015).

2) May performed mutations contribute in explaining functional variation due to known polymorphisms of ABC transporters? 

This is an interesting question but there are no SNPs in the area we have studied that have anywhere near enough data on them. We have commented in the paper that this is the case.

3) Fig. 1 A. Please indicate the relationships of the transporter with the membrane and the inner/outer sides of the transporter; non expert readers will benefit from such drawing.

We have amended Figure 1A to show this for the non-expert reader

4) E285K seems to have transport capacity better than WT. This data should be commented. Which could be the explanation? What about ATP hydrolysis property of this mutant? 

We thank the reviewer for this comment as well. The ATPase measurements were made in cell membranes and so contain a significant number of other ATPases which will all contribute to the signal measured. They are the principal reason why different membrane preps show different levels of Pi release. We can identify the ABCG2 specific component by using Ko143 to inhibit ABCG2 (at the concentration we use Ko143 is ABCG2 specific). This enables us to determine a catalytically inactive mutant very easily as there is no ABCG2-specific Ko143 inhibition. However, for mutants were elevated activity is suspected this is trickier due to the high and variable background ATPases, and the inability to further enhance this with ABCG2 specific substrates. To study E285K ATPase activity would require us to purify the protein and currently the SMALP technique we used for purification does not allow ATPase activity due to its acute Mg sensitivity (see e.g. https://onlinelibrary.wiley.com/doi/full/10.1002/anie.201610778 for discussion of this). We are trying to circumvent this so that we can determine E285K's ATPase activity but currently this is not yet possible.

Round 2

Reviewer 1 Report

The amended version of the manuscript I read offers no changes that could help me re-evaluate my previous opinion. In that occasion I expressed my concerns that the study is purely descriptive of the effects of just a handful of mutations on the ABCG2 transporter, with no offer of mechanistic or physiological insights. This referee understands very well the efforts necessary to provide that information and the need to maximise the outcome of those. However, the intention of having mechanistic data published in a future article does not make this manuscript less incomplete. I welcome the reference the authors make to Cox et al. Biochem J (2018) 475 (9): 1553–1567, where they describe some other mutants in this same transporter using techniques an approaches identical to those shown in the present study. The authors mention it as a second source of mutants for that putative future article. I do not want to enter into polemics here, but I must point that that report is a good example of what I was expecting. It does not make use of any different approaches to those reported in the present manuscript, true, but the amount of mutants described and the nature of them (ca 30 different sites in the whole molecule) provided enough information to propose surface binding sites for mitoxantrone, among other conclusions. That makes it, in my opinion, publishable.
Having said all this, I regretfully need to maintain my recommendation for reject.